# Dimerization of kringle 1 domain from hepatocyte growth factor/scatter factor provides a potent MET receptor agonist

Giovanni de Nola[1],*, Bérénice Leclercq[2],*, Alexandra Mougel[2], Solenne Taront[3], Claire Simonneau[4], Federico Forneris[5], Eric Adriaenssens[6], Hervé Drobecq[2], Luisa Iamele[1], Laurent Dubuquoy[3], Oleg Melnyk[2], Ermanno Gherardi[1], Hugo de Jonge[1], Jérôme Vicogne[2]

**Hepatocyte growth factor/scatter factor (HGF/SF) and its cognate receptor MET play several essential roles in embryogenesis and regeneration in postnatal life of epithelial organs such as the liver, kidney, lung, and pancreas, prompting a strong interest in harnessing HGF/SF-MET signalling for regeneration of epithelial organs after acute or chronic damage. The limited stability and tissue diffusion of native HGF/SF, however, which reflect the tightly controlled, local mechanism of action of the morphogen, have led to a major search of HGF/SF mimics for therapy. In this work, we describe the rational design, production, and characterization of K1K1, a novel minimal MET agonist consisting of two copies of the kringle 1 domain of HGF/SF in tandem orientation. K1K1 is highly stable and displays biological activities equivalent or superior to native HGF/SF in a variety of in vitro assay systems and in a mouse model of liver disease. These data suggest that this engineered ligand may find wide applications in acute and chronic diseases of the liver and other epithelial organs dependent of MET activation.**

## Introduction

Hepatocyte growth factor/scatter factor (HGF/SF) was simultaneously and independently discovered as a hepatotropic factor in the serum of hepatectomized rats (Nakamura et al, 1984) and as a SF inducing morphological changes and epithelial cells migration in medium conditioned by human embryo fibroblasts (Stoker &

Perryman, 1985; Stoker et al, 1987). After the discovery of its cognate receptor, the tyrosine kinase receptor MET (Bottaro et al, 1991), important studies using targeted gene disruption or blocking antibodies demonstrated embryonic developmental defects in the placenta (Uehara et al, 1995), liver (Schmidt et al, 1995), skeletal muscles of the limb and diaphragm (Bladt et al, 1995), and kidney (Santos et al, 1994; Woolf et al, 1995). Collectively, these studies demonstrate an early and wide involvement of HGF/SF-MET signalling in organ development.

In postnatal life, the plasma level of HGF/SF rises immediately after injury of several organs such as the liver, kidney, and heart (Lindroos et al, 1991; Matsumoto & Nakamura, 1997; Nakamura et al, 2000; Zhu et al, 2000). HGF/SF is essential for the regeneration the of kidney (Zhou et al, 2013), liver (Borowiak et al, 2004), and skin (Chmielowiec et al, 2007), and importantly, treatments with recombinant HGF/SF prevents fibrosis after experimental injury in the kidney (Liu, 2004) and lung (Mizuno et al, 2005), providing a strong rationale for harnessing HGF/SF-MET signalling for regenerative medicine.

The mature, biologically active species of HGF/SF is a two-chain ($\alpha/\beta$), disulphide-linked protein structurally related to the blood proteinase plasminogen. The $\alpha$-chain comprises an N-terminal (N) domain homologous to the plasminogen activation peptide and four kringle (K) domains (64 kD) and contains the high affinity-binding site for MET. The $\beta$-chain (34 kD) is homologous to the catalytic domain of serine proteinases (SPH) (Fig 1A) (Donate et al, 1994); it is devoid of enzymatic activity but contains a secondary binding site for MET. During tissue remodelling and at sites of tissue damage, the activation of the clotting/fibrinolytic pathways promotes the cleavage of the single chain pro-HGF/SF into mature

[1]Department of Molecular Medicine, University of Pavia, Unit of Immunology and General Pathology Section, Pavia, Italy  [2]University of Lille, CNRS, Inserm, CHU Lille, Institut Pasteur de Lille, U1019, UMR 9017, CIIL, Center for Infection and Immunity of Lille, Lille, France  [3]University of Lille, Inserm, CHU Lille, U1286, INFINITE, Institute for Translational Research in Inflammation, Lille, France  [4]Roche Pharmaceutical Research and Early Development (pRED), Pharmaceutical Sciences, Roche Innovation Center Basel, Basel, Switzerland  [5]The Armenise-Harvard Laboratory of Structural Biology, Department of Biology and Biotechnology, University of Pavia, Pavia, Italy  [6]University of Lille, CNRS, INSERM, CHU Lille, Centre Oscar Lambret, UMR 9020, UMR 1277, Canther, Cancer Heterogeneity, Plasticity and Resistance to Therapies, Lille, France

Correspondence: jerome.vicogne@ibl.cnrs.fr; Co-corresponding author: hugo.dejonge@unipv.it
Giovanni de Nola's present address is Department of Neurobiology, Harvard Medical School, Boston, MA, USA.
Bérénice Leclercq's present address is SATT Nord, Lille, France.
*Giovanni de Nola and Bérénice Leclercq contributed equally to this work.

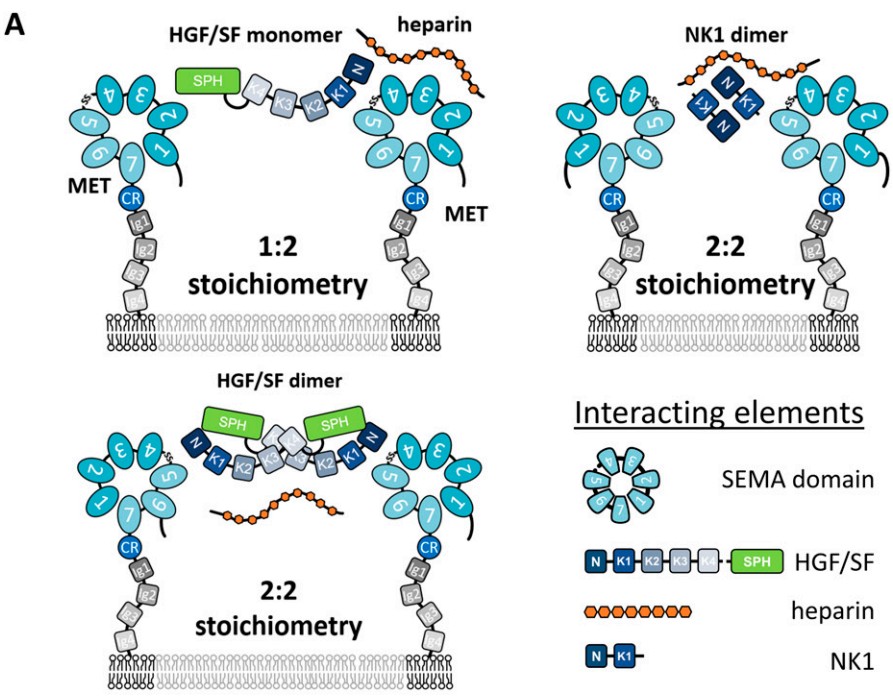

**Figure 1. Potential receptor binding and activation modes, domain architecture of MET agonists, and surface representation of NK1 crystal structure.**
**(A)** Possible receptor-binding modes and stoichiometries for hepatocyte growth factor/scatter factor and NK1 in the presence of heparin.
**(B)** Schematic representation of hepatocyte growth factor/scatter factor, NK1, K1K1, K1K1 variants, K1H6, full-length c-MET, and the MET567 fragment. Individual domains (boxes) with positions of domain boundaries indicated above. CR, cysteine rich; Ig, immunoglobulin-like; SPH, serine protease homology; TK, tyrosine kinase. The transmembrane domain is indicated in red.

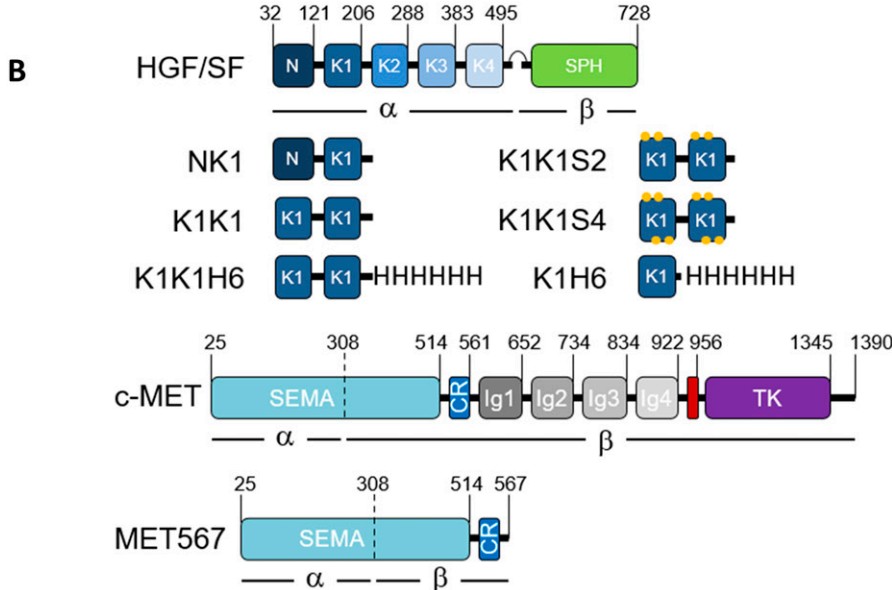

two-chain HGF/SF, leading to MET activation. In this way, HGF/SF-MET signalling promotes cell survival, division, migration, and a complex morphogenetic program underlying tissue regeneration (Miyazawa et al, 1994, 1996).

The discovery of two alternative HGF/SF splice variants, NK1 and NK2 (Chan et al, 1991; Miyazawa et al, 1991; Cioce et al, 1996), enabled further progress toward the identification of regions important for receptor binding and activation (Hartmann et al, 1992; Lokker et al, 1992). NK1, a partial receptor agonist containing the N-domain and first kringle domain (K1), is monomeric in solution but assembles as a head-to-tail non-covalent dimer in crystal structures offering an attractive bivalent receptor-activating configuration (Ultsch et al, 1998; Chirgadze et al, 1999). Heparin and heparan sulphate proteoglycans cause dimerization of NK1 in solution (Schwall et al, 1996) and are essential for the partial agonistic activity of NK1 but not for the activity of HGF/SF (Zioncheck et al, 1995; Schwall et al, 1996; Sakata et al, 1997), and on the strength of the available biochemical data and partial crystal structures, alternative models of HGF/SF-MET activation were proposed involving with 1:2 or 2:2 stoichiometries (Fig 1A) (Donate et al, 1994; Chirgadze et al, 1998; Miller & Leonard, 1998).

A recent cryo-EM structure of HGF/SF in the complex with the extracellular domain of MET (MET927) fused to a dimerization motif has offered new insights into the binding mechanisms (Uchikawa et al, 2021). This structure unambiguously shows a single HGF/SF molecule bridging two MET receptors by engaging both the NK1- and SPH-binding sites with a second HGF/SF molecule and heparin stabilizing the interaction. In contrast, a second structure of the NK1-MET927 complex confirmed the presence of the complex with 2:2 stoichiometry postulated on the basis of the early crystal structures of NK1. This structure validates independent studies on the N and K1 domain of HGF/SF produced in yeast (Holmes et al, 2007) or by total chemical synthesis (Raibaut et al, 2013; Simonneau et al, 2015), suggesting K1 domain as the main or sole domain of HGF/SF responsible for the primary, high-affinity interaction between NK1 or HGF/SF and the SEMA domain of MET.

On the strength of the structural, biochemical, and mutagenesis data available, we rationalized that a covalent dimer containing a tandem repeat of the K1 domain of HGF/SF could constitute a minimal and stable MET agonist able to diffuse effectively across tissue boundaries owing to the absence of the N-terminal domain of HGF/SF responsible for high-affinity binding to heparan sulphate proteoglycans (HSPG). Here, we report the development of this new MET ligand which we name K1K1. We describe its structure and demonstrate that K1K1 is stable and displays biological activity, equivalent or superior compared with HGF/SF, in both in vitro and in vivo models. This new MET ligand may constitute a major step towards the prospect of harnessing MET signalling for the therapy of acute and chronic degenerative diseases of major epithelial organs.

## Results

### Rational for the design of K1K1

The molecular architecture of full MET receptor, its extracellular fragment, and the various agonists used in this study are presented in Fig 1B. The concept of connecting two HGF/SF kringle 1 domains in tandem for the creation of the potent MET agonists described in this work comes from the initial observation that NK1 monomers form dimers upon crystallization providing two MET receptor–binding sites at opposite ends (Ultsch et al, 1998; Chirgadze et al, 1999). Within the structure of the NK1 dimer, the N- and C-termini of both kringle domains are all located in close proximity at the centre of the dimer structure, suggesting that a cross-linker could be made to link two kringle domains for mimicking the arrangement of K1 domains in the NK1 dimeric complex. The recreation of such a configuration was first achieved using the streptavidin molecular platform combined with C-terminally biotinylatined synthetic kringle 1 (K1B) (Simonneau et al, 2015). Oligomerization would place the K1B C-termini close together (~2 nm) around the streptavidin core and project the K1-MET–binding site located around residue Glu159 away from it, mimicking the arrangement of K1 domains in NK1 crystal structures (Fig 2). The remarkable potency of streptavidin-K1B complex confirmed our assumption that multimerization would lead to strong receptor activation and thus completed earlier observations for the location of the high-affinity

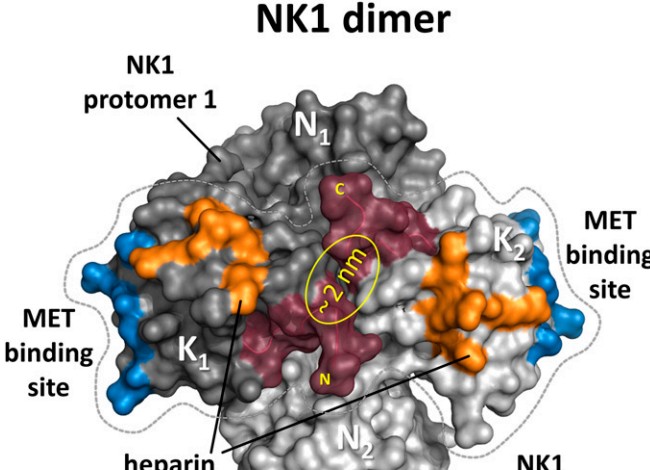

**Figure 2. Surface representation of the NK1 dimer.**
The two protomers are shown in light and dark grey with residues involved in MET-binding and heparin-binding shown in blue and orange, respectively. The centrally placed N- and C-terminus that were connected by a linker are shown in Bordeaux.

receptor-binding site in the first kringle domain (Lokker et al, 1994; Chirgadze et al, 1999; Gherardi et al, 2006). As potential heterogeneity of the streptavidin-K1B complex and the streptavidin moiety itself precluded its structural study and development as a therapeutic, we hoped that strong agonistic activity could be achieved as well by associating two K1 domains, in tandem and covalently (Fig 2). Importantly, the activity of a such a covalent K1 dimer would no longer rely on a protein dimerization or multimerization assembly as required for wild-type NK1 (Schwall et al, 1996; Sakata et al, 1997) and streptavidin-K1B.

In wild-type NK1, the distance between the last cysteine in one kringle and the first cysteine in the other kringle is roughly 1 nm, giving possibility to associate the two kringle 1 domains in tandem. An asset of such a design is to enable the production of the molecule by recombinant methods. Therefore, we designed a linker to bridge the distance between the C-terminus of the first K1 domain and the N-terminus of the second one (Fig 3A). We chose for that purpose the four–amino acid long linker, SEVE, which is naturally present between kringle 1 and kringle 2 (K2) in HGF/SF. This native linker showed no contacts with neighbouring kringle domains in wild-type NK2 reported by Tolbert and colleagues (PDB code 3HN4) (Tolbert et al, 2010) and in NK2 in complex with heparin (PDB code 3SP8) and was therefore believed to not constrain the relative orientation of the K1 domains. Furthermore, a poly-histidine–tagged variant, designated K1K1H6, was produced to facilitate certain assays and a single kringle 1 domain poly-histidine–tagged variant (K1H6) as a monovalent control. As earlier studies had revealed a secondary low-affinity heparin-binding site in the kringle 1 domain affecting biological activity (Lietha et al, 2001), we designed two additional variants, K1K1S2 and K1K1S4. In these variants, we have introduced two or four reverse-charge mutations in each kringle (schematically shown in Fig 1A) to test the implication of the basic residues naturally present at those

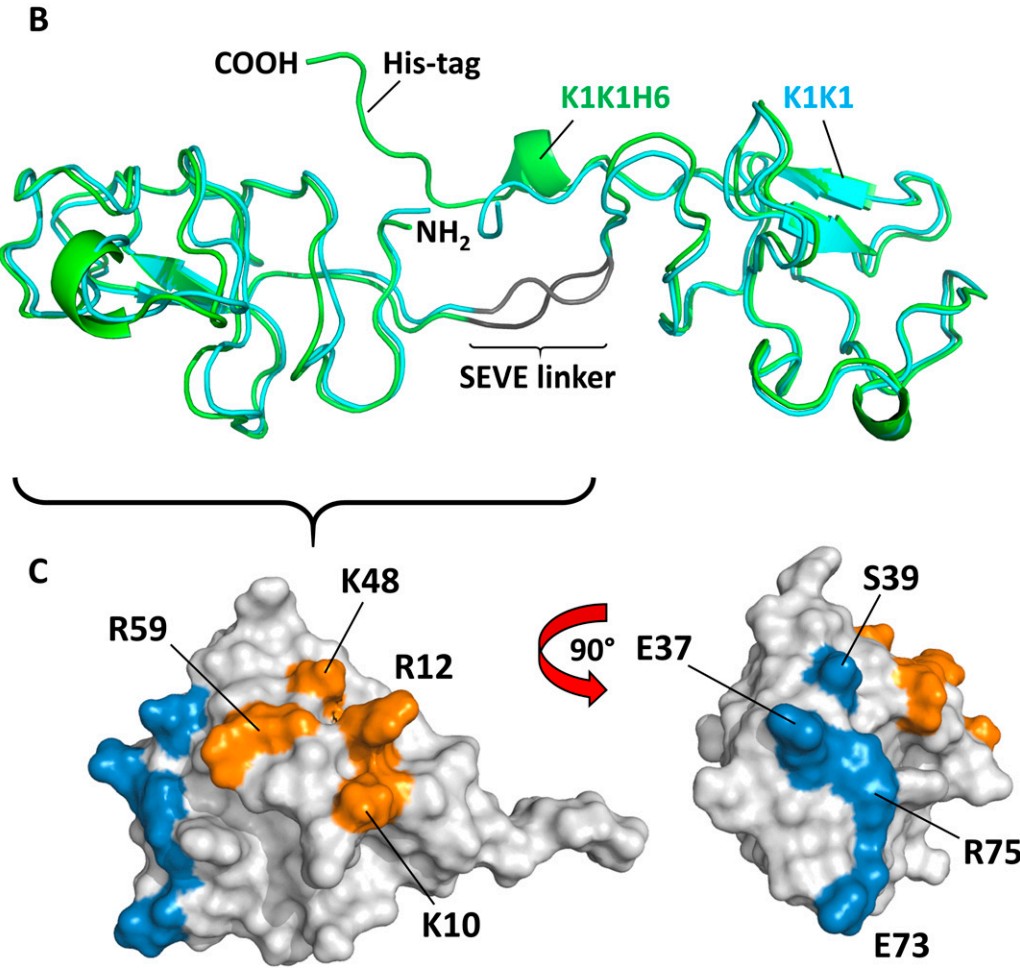

**Figure 3. Amino acid sequence, overall structure, and location of binding sites.**
**(A)** Amino acid sequence of K1K1H6 with SEVE linker in dark grey and poly-histidine tag in green. **(B)** The crystal structure of the two molecules of K1K1 (cyan) and K1K1H6 (green) showing the straight conformation and the N-terminus (NH₂) and C-terminus (COOH) located centrally in the linker region and a nearly identical overall structure with a root mean square deviation ranging from 0.8 to 1.8 Å. The C-terminal poly-histidine tag of K1K1H6 is making contacts with residues in the N-terminal kringle domain. **(C)** Surface representations of single kringle domains of K1K1 showing the location of the residues involved in MET binding, as defined by Lokker et al (1994) 40, shown in blue. The more lateral position of the heparin-binding site is shown in orange with the residues targeted by reverse-charge mutation in K1K1S2 (K10E, R12E) and K1K1S4 (K10E, R12E, K48E, R59E) indicated.

positions and to evaluate their importance for the biological activity.

Upon expression in *Escherichia coli*, K1K1 was abundantly present in inclusion bodies. Expression at 18°C and induction with a low IPTG concentration (0.1 mM) allowed the production and extraction of fully folded protein from "non-classical" inclusion bodies using a mild, non-denaturing arginine-based extraction method derived from the work of Jevsevar et al (2005). Expression in

these inclusion bodies offers an inexpensive and abundant source of recombinant protein with relatively few contaminants as opposed to soluble bacterial expression. The low-affinity heparin-binding site in the kringle domain still allowed an effective single-step purification by heparin Sepharose affinity chromatography after arginine extraction (Fig S1A). An additional size-exclusion chromatography step was used to remove possible traces of aggregation to yield crystallography-grade material with UPLC-MS

analysis confirming the purity and identity of the proteins as shown in Fig S1B and C for K1K1 and K1K1H6, respectively. Fig S1D shows all purified recombinant proteins used in this study on Coomassie-stained gel under reducing conditions.

## Structure determination by X-ray diffraction and SAXS

Purified proteins were used for crystallization experiments using commercial sparse matrix screens and several conditions resulted in the growth of protein crystals at 17°C within days. The collected datasets enabled the determination of the molecular structures at 1.8 Å resolution and 1.7 Å for K1K1 and K1K1H6, respectively. Both proteins crystalized in space group P 1 21 1 with K1K1 having two molecules per asymmetric unit and K1K1H6 having one.

Structurally, K1K1 and K1K1H6 are nearly identical with a backbone root mean square deviation of 0.8 or 1.8 Å (Fig 3B), depending on which of the two molecules in the asymmetric unit of K1K1 is used for alignment with K1K1H6. Both proteins show an elongated "stretched-out" conformation with the MET-binding sites exposed on opposite ends and the N- and C-terminus located at the centre (Fig 3C). In K1K1H6, good electron density was also observed for the poly-histidine tag possibly because of stabilising interactions between the histidine residues and residues of the N-terminal kringle domain.

Surprisingly, like the revelation of seeing the first structures of NK1 that presented a likely biologically relevant dimer, both crystal structures show a pseudo C2 symmetry around a central axis and present both receptor-binding sites in nearly identical orientation on either side of the molecule (located around residue Glu37 in the first kringle and residues Glu120 in the second kringle, equivalent to NK1 residue Glu159). One can envision a binding mode in which two MET receptor monomers are brought in proximity by binding on opposite sides of K1K1, forming a 2:1 receptor activation complex. K1K1 therefore presents the most minimalistic peptide-based receptor agonist mimicking the binding and receptor activation mechanism proposed for wild-type NK1 but with the important difference that ligand dimerization is not required. This covalent "mimicry" is even more evident when comparing the structures of the NK1 dimer and K1K1. Alignment of the first kringle domain of K1K1 with one of the two kringle domains within the NK1 dimer places the second K1K1 kringle close to the position of the kringle domain of NK1 protomer 2 (Fig S2A). However, the straightened conformation observed in the crystal structure of K1K1 "misaligns" the second K1K1 kringle domain through a rotation by 109.7 degrees and a translation of roughly 14 Å (Fig S2B). No contacts between the two K1 domains or the kringle domains with the SEVE linker are observed. The linker is more stretched out compared with the two NK2 structures supporting our idea that it is highly flexible and therefore allows significant conformational freedom (Fig S2C). This flexibility was confirmed with SAXS measurements of K1K1 in solution which generated an envelope that perfectly accommodates K1K1 volume-wise but only in a bent conformation (Fig 4A). This is also evident from comparing the experimental scatter curve with the one generated based on the K1K1 crystal structure using CRYSOL, indicating that the overall volume is the same but the distribution and therefore conformation differs significantly. Notably, the K1K1 solution SAXS envelope is compatible with the two

kringle domains in the orientation found within the NK1 dimer (Fig S2D). Excitingly, SAXS measurements of K1K1 in complex with MET567 gave a molecular mass estimation of 86.8 kD and a calculated envelope that corresponds to a 1:1 complex with a "pan handle" extrusion which could accommodate K1K1 bound to the SEMA domain (Fig 4B). Using a low-resolution crystallographic dataset of NK1 in complex with MET567 (unpublished, briefly described in Blaszczyk et al [2015]), we could generate a model based on the alignment of K1K1 with the crystal structure of the NK1-MET567 complex. The theoretical scatter curve matched the experimental scatter curve remarkably well ($\chi^2$ = 3.6) with also in this model a slightly bent K1K1 conformation fitting best with the SAXS envelope. Moreover, the recently published cryo-EM structure of the NK1 dimer in complex with two SEMA domains (Uchikawa et al, 2021) perfectly accommodates our K1K1 SAXS envelope overlapping both NK1 kringle domains and supports the proposed model for receptor activation by K1K1 (Figs 4C and S2E). Overall, we have generated compelling structural support for a minimal MET receptor agonist that comprises two covalently linked MET-binding sites. The structure of K1K1 and K1K1H6 are available from the PDB data bank with id code 7OCL and 7OCM, respectively. The SAXS data are available from SASBDB database with id codes SASDLM9, SASDLN9, and SASDLP9.

## In vitro biological data

To analyse MET receptor activation and downstream signalling induced by K1K1, HeLa cells were stimulated with K1K1, K1K1H6, and monovalent K1H6 at different concentrations using human HGF/SF as a reference. MET receptor phosphorylation and activation of the key downstream signalling molecules Akt and ERK were confirmed by Western blot (Fig 5A) in which both K1K1 and K1K1H6 stimulate phosphorylation of the pathway down to the lowest tested concentration of 100 pM. As predicted, treatment with K1H6 is indistinguishable from the negative control treatment, like what we observed with the chemically synthesized K1 domain previously (Ollivier et al, 2012). We then performed precise quantification of Akt and ERK activation in HeLa cells stimulated with HGF/SF, K1K1, K1K1H6, and K1H6 using AlphaScreen technology. For p-Akt, both K1K1H6 and K1K1 show similar phosphorylation in the low nanomolar range with no significant effect caused by the presence or absence of the poly-histidine tag. Compared with HGF/SF, both variants show lower maximum activation values of Akt (Fig 5B). Interestingly, activation of the Ras-Raf-MAPK pathway as measured by the phosphorylation of ERK showed no significant differences between stimulation by HGF/SF, K1K1H6, or K1K1 at the tested concentrations (Fig 5C). Again, no activity was observed with K1H6 even at the highest concentrations. We next confirmed the direct binding of K1K1H6 to recombinant human MET-Fc chimera, a soluble dimeric protein comprising the whole extracellular part of the MET receptor (MET922 fused to the Fc region of human IgG1) at different concentrations (Figs 5D and S3A) using the AlphaScreen technology. To accurately determine the $K_D$, we performed a full kinetic analysis of K1K1 binding to immobilized MET567 by surface plasmon resonance (SPR) spectroscopy which furnished an apparent $K_D$ of 205 nM (Fig S3B), which is similar to the $K_D$ obtained for NK1 (200 nM, Fig S3C).

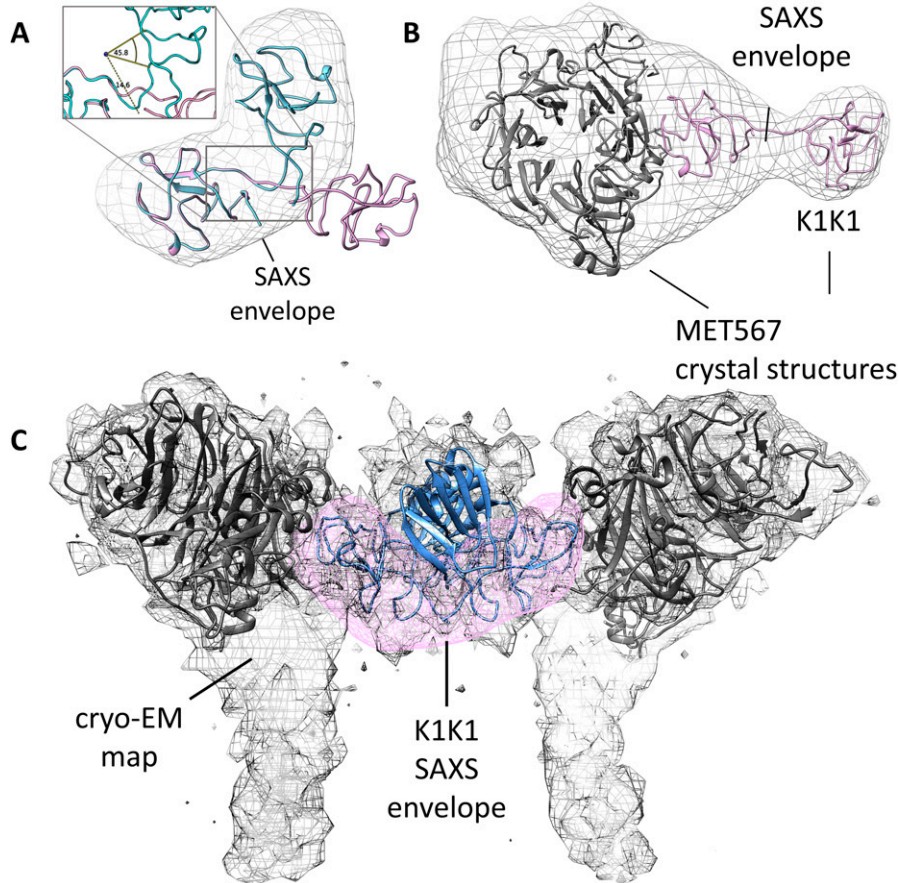

**Figure 4.    SAXS models for K1K1 alone and in complex with MET567 receptor fragment.**
**(A)** The measured SAXS envelope is not compatible with the elongated K1K1 crystal structure (pink) but perfectly accommodates a bend conformation of K1K1 (cyan). To fit the SAXS envelope, the linker region is bent by roughly 60°. **(B)** The ab initio SAXS envelope of the 1:1 complex of K1K1, and MET567 receptor fragment shows a "pan-handle" extension, which accommodates K1K1. The CR domain is partially protruding from the bottom of the envelope, and extra volume might be occupied by glycosylated side chains not modelled. Images generated with UCSF Chimera. **(C)** The SAXS envelope of K1K1 in solution (pink) overlaps well with part of the cryo-EM map (emd_23923.map) accommodating the two kringle domains of the NK1 dimer (blue). Cryo-EM map for 7mob.pdb.

Having established MET receptor binding and remarkably potent activation of the MET signalling pathway by both K1K1 and K1K1H6, we subsequently focused on demonstrating its potency in various phenotypic biological assays. We first performed a cell viability assay based on metabolic activity on MDCK cells incubated overnight (16 h) in the presence of anisomycin, an apoptosis inducer, which prevents protein synthesis and leads to cell death. We looked at the cell survival after the addition of HGF/SF, K1K1, K1K1H6, and NK1 at different concentrations during anisomycin treatment. Although HGF/SF was most effective, K1K1 and K1K1H6 were equals in preventing cell death and much more potent than native NK1 in this assay (Fig 5E). We also used this assay to validate the specific activities of different preparations of K1K1 and K1K1H6 showing very little batch-to-batch variation and similar relative activity when compared with HGF/SF and NK1 (Fig S4A), confirming the reliability of the protein production and purification protocol.

An MDCK scatter assay (Stoker & Perryman, 1985) was used as a sensitive phenotypical assay to determine the minimal concentration needed to activate MET and induce the epithelial–mesenchymal transition and cell scattering. K1K1 was able to exert an effect at a concentration down to 1 pM, 10 times lower than native HGF/SF and 1,000 times lower than NK1 (Fig S4B).

To examine the role of the low-affinity heparin-binding sites in the kringle domain, we used the analogues K1K1S2 and K1K1S4 with specific reverse-charge mutations altering heparin binding. These variants were compared with HGF/SF and K1K1 in an MDCK scattering assay (Fig 6A). Both variants had reduced activity with observable scattering for K1K1S2 down to 300 pM and no scattering observed at 100 pM, whereas K1K1 still showed activity at a concentration 10 times lower (10 pM). K1K1S4 only showed activity in the high nanomolar range and was inactive at 1 nM. We further performed a 3D morphogenesis assay using MDCK cells and observed branched structures induced by low nanomolar concentration of HGF/SF and K1K1 at which no branching was observed for K1K1S and K1K1S4 (Figs 6B and S5A and B). We confirmed the results obtained with the scatter assay, indicating that the reverse-charge mutations reduced the biological activity with a net reduction of tubulogenesis with K1K1S2 and a complete absence with K1K1S4 at 10 nM. The loss of biological activity induced by the mutations was also observed in SKOV3 cells using a quantitative Boyden chamber migration assay (Fig 6C), showing a significant difference of activity between K1K1 and the mutant variants. Finally, we examined the impact of the mutations on biological activity at the level of the MET receptor signalling pathway by investigating the phosphorylation status of MET, Akt, and ERK (Fig S5C). Altogether, these data indicate that K1K1S2 and K1K1S4 are weaker agonists than K1K1 by one and two orders of magnitude, respectively, and suggest that the low-affinity heparin-binding site within the K1 domain plays a significant role in MET activation by K1K1.

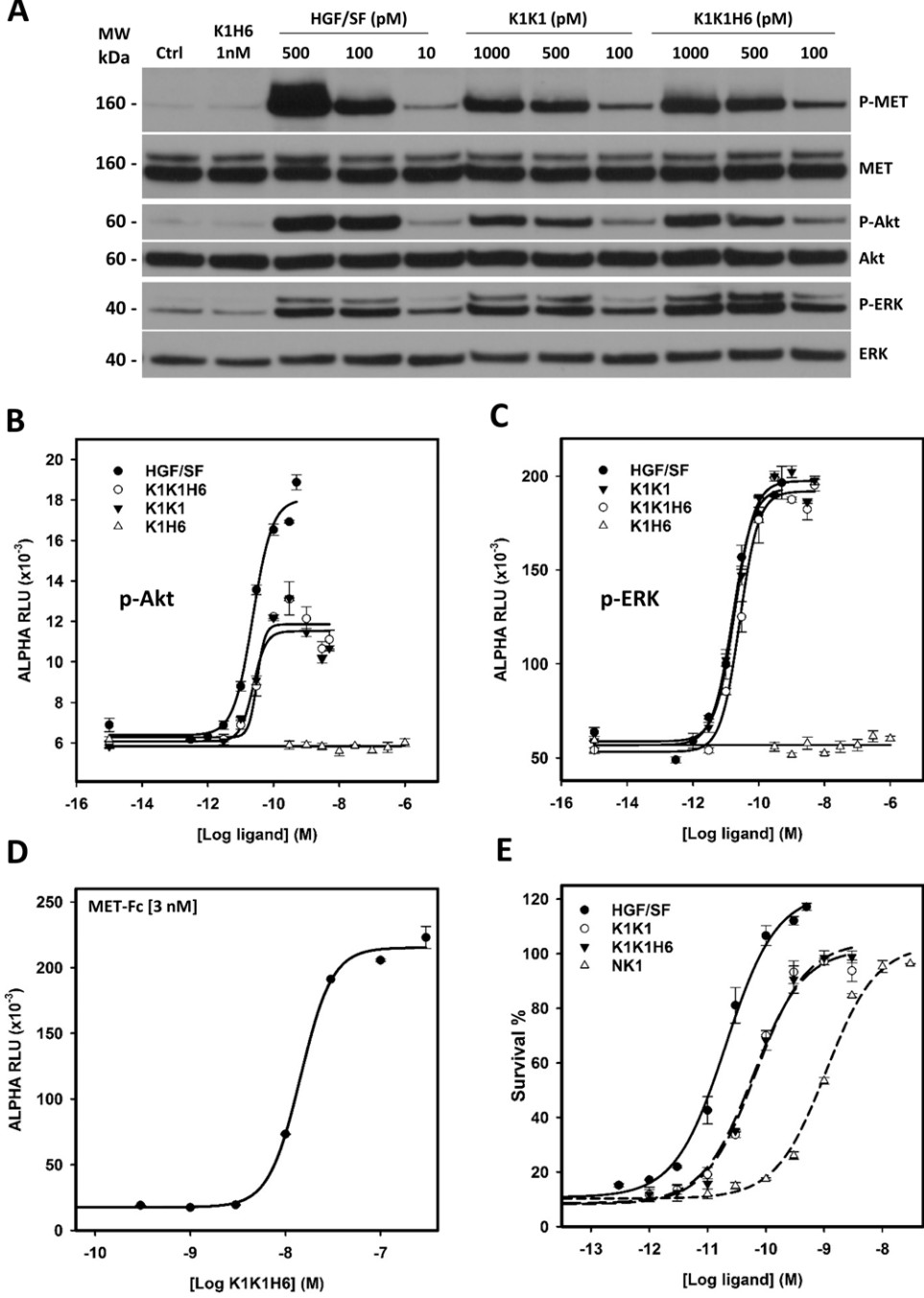

**Figure 5. In vitro activity of the hepatocyte growth factor/scatter factor (HGF/SF), K1K1, K1K1H6, and K1H6.**
**(A)** Phosphorylation analysis by Western blot on Hela cell lysates after stimulation with ligands for 10 min at concentrations indicated above each lane. Loading controls are based on total MET, total Akt, and total ERK present in each lane. **(B, C)** AlphaScreen measurements of p-Akt and (C) p-ERK activation in HeLa cells after 10 min stimulation with K1K1, K1K1H6, HGF/SF, and K1H6. **(D)** Binding determination using the AlphaScreen saturation binding assay. Seven concentrations of K1K1H6 were tested on several different concentrations of MET-Fc (AA 25–922) (Fig S3A). Shown is the binding of K1K1H6 to 3 nM MET-Fc. **(E)** Ligand induced MDCK survival after overnight treatment with the apoptotic inducer anisomycin. Indicated is the percentage of viable cells compared with no-anisomycin treatment after exposure to HGF/SF, K1K1, K1K1H6, and NK1 at different concentrations.
Source data are available for this figure.

## In vivo activation of MET signalling

Having established the superior potency of K1K1 over NK1 in different in vitro assays that sometimes matched or even surpassed native HGF/SF, we were keen to start several in vivo mouse studies to look at the effects of K1K1. We first validated that K1K1 is diffusible and biologically active in vivo, whatever the injection route (I.P. versus I.V.), by measuring phosphorylation of the MET receptor, Akt, and ERK in liver homogenates 10 min after injection (Fig S6). Because no significant difference was observed between I.P. and I.V.

administration routes, all subsequent injections were done by I.P for experimental convenience. To study the effect of different doses, 8-wk-old FVB mice were injected with different amounts of K1K1 (0.1–5 μg per 20 g animal) after which the animal was euthanized 10 min later. The phosphorylation status of the MET signalling pathway was established in the liver homogenates. Control and 0.1 μg of K1K1 did not result in any detectable MET, Akt, and ERK phosphorylation signal compared with control, whereas injection of 0.5, 1, and 5 μg gave uniform activation of the pathways (Fig 7A). A second experiment was performed to determine the

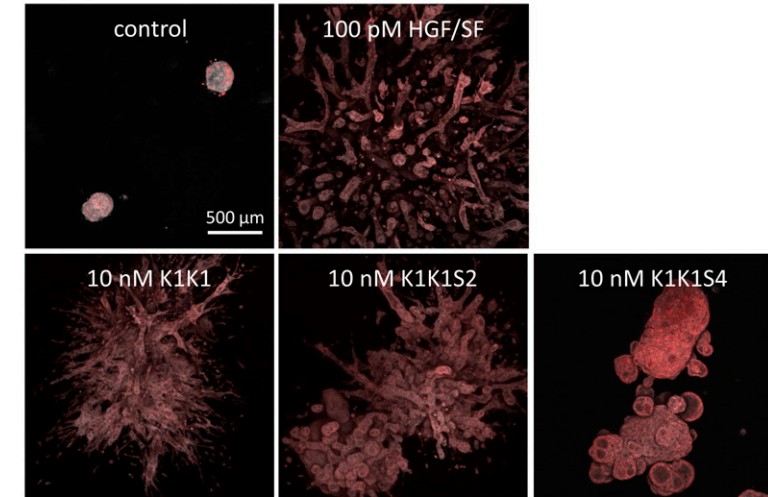

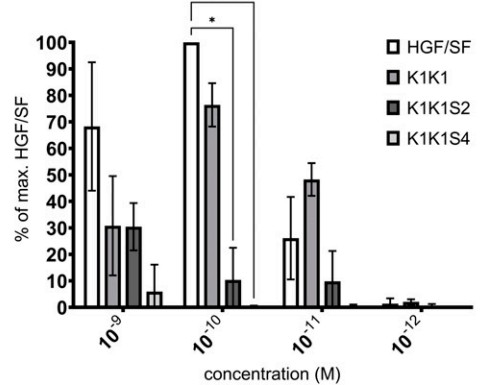

**Figure 6. Effects on cell motility, migration, and morphology.**
**(A)** MDCK cell scattering at different concentrations K1K1, K1K1S2, and K1K1S4 showing the lowest concentration at which each protein is still active and the subsequent dilution at which no more scattering is observed. Hepatocyte growth factor/scatter factor (HGF/SF) is used as positive control to generate maximum scattering. For complete half-log dilution of agonist series, see Data Source Image File. **(B)** 3D reconstruction by z-stacking of fluorescence microscopy images taken of large MDCK cell colonies stimulated with 100 pM HGF/SF or 10 nM K1K1 and mutants for 4 wk. The combined fluorescence of DAPI (red) and Evans blue staining (grey scale) shows the extensive branching morphogenesis and tubulogenesis induced by both proteins. **(C)** Boyden chamber migration assay using SKOV3 cells. Three independent assays were performed in which SKOV3 cells were treated for 6 h with indicated concentrations of HGF/SF, K1K1, K1K1S2, and K1K1S4. 0.1 nM HGF/SF was taken as 100% to which all other conditions where compared. Statistical significance was calculated with ANOVA followed by Dunnett's test with $P < 0.05$ and * indicates significantly difference. Error bars represent mean ± SD based (n = 5).
Source data are available for this figure.

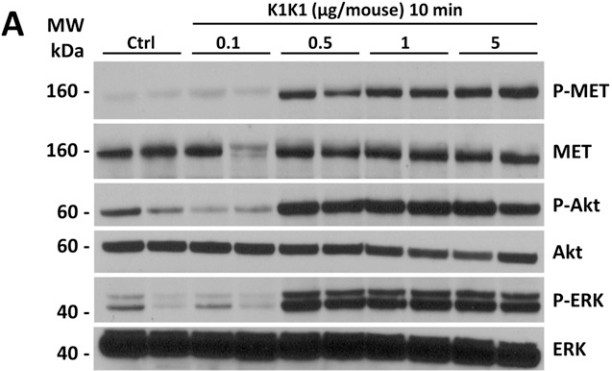

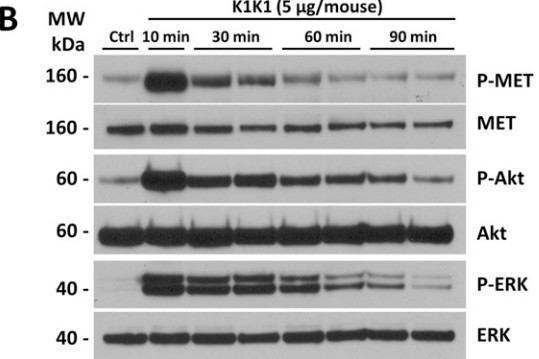

**Figure 7.  Dose response and MET pathway activation kinetics after i.p. injection.**
**(A)** 8-wk-old FVB mice were injected with PBS (Ctrl) or different amounts of K1K1 after which MET, AKT, and ERK phosphorylation in liver homogenate was determined using Western blot. Mice were euthanized 10 min after injection. Results are presented as experimental duplicate (n = 2). **(B)** MET, AKT, and ERK phosphorylation were detected by Western blot at different time points after injection of 5 μg of K1K1. Results are presented as experimental duplicate (n = 2) except for control and 10 min conditions. Both blots present total MET, Akt, and ERK proteins as loading controls.
Source data are available for this figure.

duration of the stimulation by euthanizing the animals at 10, 30, 60, and 90 min, using a K1K1 dose of 5 μg. The data obtained show a diminishing signal over time detectable up to 60 min (Fig 7B) that could correspond to natural receptor desensitization rather than K1K1 protein half-life.

### In vivo efficacy of K1K1 in the treatment of alcoholic steatohepatitis

To evaluate the in vivo efficacy of K1K1 applied to a liver disease model, we used a validated mouse model of subchronic alcohol exposure, the adapted Lieber DeCarli (LDC) model (Lieber & Decarli, 1989) to study steatosis, a common feature of several liver diseases.

As expected, alcohol consumption induced steatosis in the liver of mice fed with alcohol (LDC + vehicle group) which was not observed in mice fed with a control diet (control + vehicle) or a control diet with K1K1 treatment (control + 10 μg K1K1) (Figs 8A and S7A). The treatment with different doses of K1K1 (0.4, 2 and 10 μg) significantly decreased the steatosis in mice fed with alcohol (LDC + K1K1) (Fig 8B). Regarding the factors involved in steatosis

improvement, K1K1 treatment was able to significantly increase mRNA expression of ApoB, PPARα, and LDLR (Fig 8C). The highest doses of K1K1 (2 and 10 μg) also significantly decreased the mRNA expression of TNFα and IL-6 in LDC mice, whereas a moderate increase of both markers was observed in untreated mice compared with control health mice. Interestingly, although mRNA expression of MET decreased by ethanol exposure alone, it was induced by K1K1 treatment (Fig S7B).

Taken together these results show that K1K1 significantly improved steatosis in a mouse model of liver injury. Our data suggest that this therapeutic effect is mediated in part by the increase of protective factors such as ApoB and PPARα and the lowering of proinflammatory cytokines, induced by K1K1 treatment.

## Discussion

HGF/SF-MET signalling is essential for liver development (Schmidt et al, 1995; Uehara et al, 1995) and regeneration (Borowiak et al, 2004; Huh et al, 2004; Liu, 2004) and pre-clinical studies employing either recombinant HGF/SF (Horiguchi et al, 2009), HGF/SF gene delivery (Ueki et al, 1999) or HGF/SF-transfected mesenchymal stem cells (Moon et al, 2019) all demonstrated a strong therapeutic potential for HGF/SF in pre-clinical models of liver diseases. However, all failed to offer a robust platform toward the use of HGF/SF in the clinic. Indeed, both gene therapy and cell-based therapies meet considerable regulatory challenges and therapeutic applications of recombinant HGF/SF face strong and well-known limitations: the protein has a complex multidomain protein susceptible to proteolytic degradation, is unstable in physiological buffers, and exhibits limited tissue diffusibility and penetration. Hence, harnessing the HGF/SF-MET pathway in regenerative medicine will rely on engineered and improved HGF/SF derivatives in terms of stability and tissue availability.

Several molecules have been developed in recent years to overcome the limitations of HGF/SF in therapy. These included engineered variants of the natural growth factor (Cassano et al, 2008; Sinha Roy et al, 2010), agonistic monoclonal antibodies (Gallo et al, 2014; Kim et al, 2019), synthetic cyclic peptides (Sato et al, 2020), and aptamers (Ueki et al, 2020). All the engineered variants of HGF/SF developed thus far, however, contain the high-affinity HSPG-binding site present in the N-domain and thus share with HGF/SF the feature of limited diffusion and tissue penetration. The design of a new MET ligand with reduced affinity for HSPG therefore was the prime objective of the present study. Building on available structures, mutagenesis data, and the biological activity of multimeric assembly of the K1 protein (Simonneau et al, 2015), we rationalized that a K1K1 covalent dimer may have yielded a MET agonist with interesting physico-chemical and biological properties.

The structural data reported in this work show that K1K1 dimer displays a perfect symmetrical positioning of the K1 units in its crystalized form and thus readily open the possibility of a direct MET dimerization in a 1:2 K1K1:MET configuration. SAXS data indicate a bending of K1K1 when in solution, nicely fitting the natural position of the K1 domains within the NK1 dimer bound to MET as observed in recent cryo-EM structural analysis (Uchikawa et al,

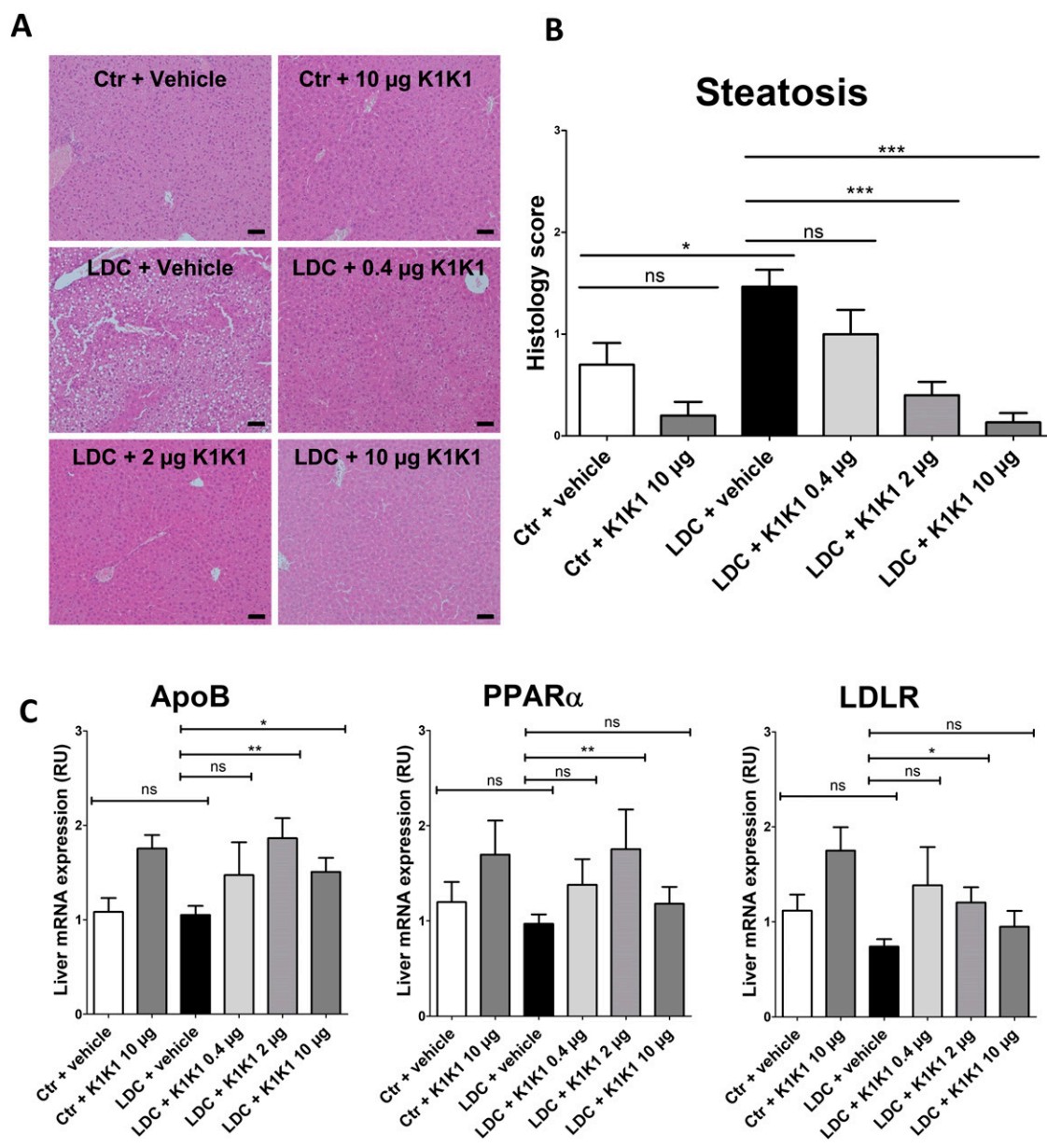

**Figure 8. In vivo evaluation of the efficacy of K1K1 in a mouse model of alcoholic steatohepatitis.**
**(A)** Hematoxylin-erythrosin B staining of mouse livers submitted to an adapted Lieber DeCarli model (20× magnification). Analyses were performed on 10 animals in control groups (Ctr + vehicle and Ctr +10 µg K1K1) and 15 animals in ethanol-treated groups (LDC ± K1K1). Scale bar = 50 µm. **(B)** Steatosis was quantified in each mouse liver using a steatosis score (see STAR Methods). **(C)** Results are expressed as mean ± SD (C). The mRNA expression of triglyceride metabolism markers (ApoB, PPARa, and LDLR) was analysed in mouse livers using RT-qPCR with b-actin as housekeeping gene. Results are expressed in relative units (RU) and represented as mean ± SD. Source data are available for this figure.

2021). Thus, the structural data suggest that flexibility is important for enabling the proper positioning of K1 domains upon binding to MET.

K1K1 displayed a vastly superior stability in physiological buffers compared with native HGF/SF and a potency approaching activity at picomolar concentrations in vitro (Figs 4A and 5A). Importantly, the improved diffusibility into tissues as a result of reduced binding to heparin (Fig S8) clearly is a further and major component of the remarkable activity displayed by K1K1 in the studies in vivo (Figs 6 and 7). SPR analysis confirmed a 10-fold reduction of heparin binding in

K1K1 ($K_D$ ~ 5.6 µM) compared with NK1 ($K_D$ ~ 0.43 µM), whereas the affinity of the K1K1 mutants S2 and S4 was below the limits of sensitivity of SPR (Fig S8). Interestingly, these mutants displayed reduced biological activity compared with K1K1 indicating that the presence of the low affinity site for heparan sulphate in K1 domain plays a role in the potent agonistic activity in vitro and in vivo. Seen from a different angle, K1K1 can be regarded as a tool that can be easily manipulated and varied to investigate more in depth the role of heparan sulphate molecules in ligand–receptor binding and receptor activation, independently of the strong HS-binding N-domain.

Altogether, the structural, biochemical, and biological data reported here suggest that K1K1 might serve as a lead candidate for protein therapy of several human pathologies where activation of the HGF/SF-MET signalling axis can be beneficial. Activation of the MET pathway has been known to ameliorate alcoholic steatohepatitis and liver fibrosis or cirrhosis in pre-clinical models of liver disease (Matsuda et al, 1995; Matsuda, 1997; Tahara et al, 1999; Taïeb et al, 2002), and conversely, deletion of the MET gene in postnatal mice causes hyperlipidemia, severe steatosis, and progressive liver disease (Kroy et al, 2014). No specific drug has been approved as yet for the treatment of alcoholic and non-alcoholic liver disease (Sharma et al, 2021), but we submit that the K1K1 protein data presented here may emerge as a strong candidate for clinical development, owing to its favourable physical chemistry and its strong activity in a well-established clinical mouse model of alcoholic steatohepatitis (Figs 7 and S7A).

Finally, we wish to emphasize that HGF/SF is a broad-acting morphogen that controls development and regeneration of other major epithelial organs beyond the liver: HGF/SF–MET signalling is essential for lung alveologenesis (Calvi et al, 2013) and for kidney development (Ishibe et al, 2009). Thus, we surmise that potent MET agonists may find many applications beyond liver disease notably in the large number of patients with acute and chronic kidney damage (Zhou et al, 2013) and chronic lung emphysema (Shigemura et al, 2005) and may well find an urgent application in reducing lung damage and fibrosis in patients suffering from COPD (Chronic obstructive pulmonary disease) or severe COVID-19 (Zuo et al, 2010; George et al, 2020).

One may argue that the HGF/SF-MET pathway is strongly involved in tumour progression and metastasis spreading. Indeed, the role of aberrant HGF/SF and MET signalling in human cancer has been extensively investigated (Gherardi et al, 2012), and the use of MET agonists in regenerative medicine clearly must meet an adequate safety profile. Results from early studies with native HGF/SF however are most encouraging because in a series of clinical trials involving a total of 5,902 patients who were administered natural HGF/SF for liver failure, no excess cancer mortality was reported (Cui et al, 2008). Thus, although further studies are clearly needed, the available data thus suggest that a safe therapeutic window for applications of MET agonists in regenerative medicine exists.

# Materials and Methods

### Cloning of K1K1 and K1K1 variants

The cDNA encoding K1K1 and K1K1H6 was generated using a fusion PCR reaction after separate PCR amplification of two K1 domains from a wild-type human NK1 template. The forward primer, ATCATCCCAT GGCCATTAGA AACTGCATCA TTGGTAAAGG ACG, was used to amplify the first kringle domain introducing a 5′ NcoI restriction site. The reverse primer for the first K1, TTCAACTTCT GAACACTGAG GA, was used to introduce the linker sequence S E V E placed in between the two K1 domains. S E V E is naturally found in HGF/SF in between the K1 and K2 domains. The forward primer for the second K1 domain, TCAGAAGTTG AATGCATCAT TGGTAAAGGA CG, introduces the

fusion-overlap with the SEVE linker sequence. A reverse primer with or without a six-histidine tag, ACAGCGGCCG CTCATCAATG ATGATGATGA TGATGTTCAA CTTCTGAACA CTGAGGA and ACAGCGGCCG CTCATCATTC AACTTCACTA CACTGAGGAA T, respectively, introduced two stop codons followed by a NotI restriction site. After the fusion PCR reaction, the product was digested with NcoI and NotI and ligated into the pET45b(+) expression plasmid (Novagen/EMD Millipore). Integration in the pET45b(+) MCS introduced the additional amino acid sequence M A I R N upstream of the first cysteine of K1 domain. The heparin mutants S2 and S4 were based on existing kringle-variant NK1 templates. These templates introduced the reverse-charge amino acid substitutions K10E, R12E, K93E, and R95E for K1K1S2 and K10E, R12E, K48E, R59E, K93E, R95E, K131E, and R142E for K1K1S4.

### Protein expression

Recombinant human NK1 protein (residues 28–209) was expressed in *Pichia pastoris* and purified as described in Chirgadze et al (1999). Recombinant human MET567 was expressed in CHO Lec 3.2.8.1 cells and purified as described in Gherardi et al (2003). Human HGF/SF was stably expressed in NS0 myeloma cells and purified as described in Gherardi et al (2006).

For the expression of K1K1 and K1K1 variants, a single colony of freshly transformed *E. coli* BL21(DE3) was used to inoculate an overnight 5 ml Luria broth (LB) culture containing 100 µg/ml of ampicillin. This culture was used to inoculate 500 ml of LB with ampicillin shaking at 250 rpm and grown to an optical density (600 nm) of 0.6–0.8 at 37°C. The culture was then cooled down to 18°C while shaking after which IPTG was added to a final concentration of 0.1 mM. The culture was grown at 18°C with shaking for 24 h after which the bacterial cells were harvested by centrifugation for 30 min at 10,000 rcf at 4°C, and the cell pellet was stored at –80°C.

### K1K1 extraction from inclusion bodies

The frozen bacterial cell pellet was thawed on ice and resuspended in 25 ml 50 mM Tris (pH 8.5), 500 mM NaCl, with the addition of one tablet of cOmplete EDTA-free Protease Inhibitor Cocktail (Roche), 1 U of Pierce Universal Nuclease and 10 µg of lysozyme. The suspension was incubated rotating at 4°C for 30 min before being placed on ice and sonicated using 10 pulses of 30 s with 60 s pause in between using a Sonic Ruptor 400 (Omni International) with a OR-T-750 tip at 100% power output. The suspension was centrifuged at 10,000 rcf for 10 min, the supernatant discarded, and the pellet resuspended in 25 ml 50 mM Tris (pH 8.5), 500 mM NaCl with 0.4% Triton X-100 using a glass Potter-Elvehjem tube and PTFE pestle. The suspension was incubated at 4°C rotating for 30 min after which it was centrifuged at 10,000 rcf for 10 min, the supernatant discarded, and the pellet resuspended in 25 ml 50 mM Tris (pH 8.5), 500 mM NaCl with 0.025% NP40 using the Potter-Elvehjem tube and pestle. The suspension was again incubated at 4°C rotating for 30 min, centrifuged for 10 min at 10,000 rcf, the supernatant discarded, and the pellet resuspended in 25 ml 50 mM Tris (pH 8.5), 500 mM NaCl using the Potter-Elvehjem tube and pestle. The suspension was once more incubated at 4°C rotating for 30 min, centrifuged at 10,000 rcf for 10 min, and the supernatant discarded. The final pellet

was resuspended in 20 ml 50 mM Tris (pH 8.5), 500 mM NaCl, 2 M arginine, 0.5 mM GSSG, 5 mM GSH. This suspension was incubated for 3 d at 4°C on a rotary wheel.

## K1K1 purification

After 3 d of incubation in Tris buffer with arginine, the inclusion body suspension was centrifuged at 20,000 rcf for 30 min at 4°C. The supernatant was transferred to a new centrifuge tube and centrifuged again at 20,000 rcf for an additional 30 min at 4°C. Unless a pellet was visible, in which case a third centrifugation was performed in a new tube, the cleared supernatant was diluted 100 times in 2 liters of 50 mM Tris (pH 7.4), 150 mM NaCl and filtered. The diluted supernatant was loaded onto a 5-ml Heparin HiTrap column (GE Healthcare) and eluted with a gradient up to 1 M NaCl, in 50 mM Tris (pH 7.4). Peak fractions were pooled, concentrated, and loaded on a HiLoad 16/600 Superdex 200 pg column (GE Healthcare) equilibrated in 50 mM Tris (pH 7.4), 500 mM NaCl. Peak fractions were pooled and concentrated to 5 mg/ml before being used or stored after flash freezing. For K1K1S2 and K1K1S4, instead of the Heparin HiTrap column, the first step purification was done on a 5-ml HisTrap HP column (GE Healthcare).

## Protein crystallization and X-ray diffraction

For protein crystallization, K1K1 and K1K1H6 were dialyzed in 10 mM Tris (pH 8.5), 100 mM NaCl and concentrated to 12 and 11.8 mg/ml, respectively. A pre-crystallization screen (PCT, Hampton Research) was performed to confirm these concentrations were favourable for crystallization after which 48-well sitting-drop plates (Hampton Research) were set up using Crystal Screen 1 (Hampton Research), JCSG-plusTM (Molecular Dimension), and Morpheus (Molecular Dimensions) crystallization screens. After overnight incubation at 17°C, the first crystals of K1K1H6 appeared in the Morpheus screen in condition C7 consisting of 100 mM MOPS/Hepes (pH 7.5), 30 mM sodium nitrate, 30 mM sodium phosphate dibasic, 30 mM ammonium sulphate, 20% vol/vol glycerol, 10% wt/vol PEG4000. Further optimization was performed with different protein to crystallization solution ratios (1:2, 1:1, 2:1) in a 24-well plate using hanging drop. After 24 h, good crystals were obtained in a 2:1 protein-to-solution ratio condition. In addition to the cryo-protection intrinsic to the Morpheus screen, additional glycerol was added up to 25% of the final volume before crystals were collected using a CryoLoop (Hampton Research) and flash frozen and stored in liquid nitrogen. Crystallizations of K1K1 went through a similar optimization procedure and yielded good crystals in a condition of 100 mM Tris/Bicine (pH 8.5), 30 mM sodium nitrate, 30 mM sodium phosphate dibasic, 30 mM ammonium sulphate, 12.5% wt/vol PEG 1000, 12.5% wt/vol PEG 3350, 12.5% vol/vol MPD. Data were collected at the ESRF in Grenoble, France, at beamline at BM14 for K1K1H6 and ID23-1 for K1K1 and was solved by molecular replacement using the NK1 kringle domain taken from 1BHT.pdb (Ultsch et al, 1998) as a search model resulting in a structure with a resolution of 1.7 Å and an Rwork/Rfree of 19.7/22.7% for K1K1 and a structure with a resolution of 1.8 Å and an Rwork/Rfree of 16.5/19.4% for K1K1H6. Both proteins were crystallized in monoclinic space group P1 21 1.

## Crystal structure determination

For both K1K1H6 and K1K1, the MTZ file generated by the ESRF EDNA framework Fast Processing System was used, downloaded through the ISPyB server. At the different session of measurement, the EDNA auto processing used AIMLESS (v0.3.3) and POINTLESS (v1.9.8) to create the MTZ file for K1K1, whereas for K1K1H6, the MTZ file was generated using AIMLESS (v0.5.2) and POINTLESS (v1.9.25). Molecular replacement was performed using Phaser (v2.8.3), the Phenix software package (v1.19.2) (Liebschner et al, 2019), using the kringle domain from 1BHT.pdb (Ultsch et al, 1998) followed by several rounds of refinement using Phenix refine (Afonine et al, 2012) and WinCoot (Emsley et al, 2010) (v0.9.4.1). The dataset of K1K1 and K1K1H6 had an upper resolution limit of 1.61 and 1.39, respectively, a resolution cut-off was set to 1.8 for K1K1 and 1.7 for K1K1H6. The data collection and refinement parameters and statistics are given in Table S1. The data collection and refinement statistics are given in Table S1. The molecular graphics presented in this manuscript were generated with PyMOL (v2.4.1) and Chimera (v1.15) (Pettersen et al, 2004).

## Small-angle X-ray scattering (SAXS)

Solution scattering data were collected at ESRF BM29 using a sec-1 frame rate on Pilatus 1 M detector located at a fixed distance of 2.87 m from the sample, allowing a global q range of 0.03–4.5 nm with a wavelength of 0.01 nm. SEC-SAXS experiments were carried out using the Nexera High Pressure Liquid/Chromatography (HPLC; Shimadzu) system connected online to SAXS sample capillary. For these experiments, 35 μl of each sample at the concentrations indicated in the Table S2 were injected into a Superdex 200 PC 3.2/300 Increase column (GE Healthcare), pre-equilibrated with 25 mM Tris (pH 7.4), 150 mM NaCl. Frames corresponding to protein peaks were identified, blank subtracted, and averaged using CHROMIXS2. Radii of gyration (Rg), molar mass estimates, and distance distribution functions P(r) were computed using PRIMUS in the ATSAS package (v3) (Petoukhov et al, 2012). Modelling of flexible loops and glycosylation were carried out using COOT, CORAL, and SASREF. Comparison of experimental SAXS data and 3D models from structural models was performed using CRYSOL. A summary of SAXS data collection and analysis results is shown in Table S2.

## AlphaScreen MET-binding assay

Saturation assays for binding of K1K1H6 to recombinant MET-Fc protein were performed in 384-well microtiter plates (OptiPlate-384, PerkinElmer, 50 μl of final reaction volume). Final concentrations were 0–300 nM for K1K1H6, 0–10 nM for hMET-Fc (8614-MT-100; R&D Systems), 10 μg/ml for Ni-NTA coated donor beads and protein A-conjugated acceptor beads. The buffer used for preparing all protein solutions and the beads suspensions was PBS (10 mM phosphate buffer [pH 7.4], 148 mM NaCl, 2 mM KCl), 5 mM Hepes [pH 7.4], 0.1% BSA. K1K1H6 (10 μl, 0–1.5 μM) was mixed with solutions of MET-Fc (10 μl, 0–50 nM). The mixture was incubated for 10 min (final volume 15 μl). Protein A-conjugated acceptor beads (#6760617C; 10 μl, 50 μg/ml; PerkinElmer) were then added to the vials. The plate

was incubated at 23°C for 30 min in a dark box. Finally, Ni-NTA coated donor beads (6760619C; 10 $\mu$l, 50 $\mu$g/ml; PerkinElmer) were added, and the plate was further incubated at 23°C for 60 min in a dark box. The emitted signal intensity was measured using standard Alpha settings on an EnSpire Multimode Plate Reader (PerkinElmer). Measurements are expressed as technical duplicates (mean ± SD, n = 2). The data corresponding to the 3 nM MET condition were subjected to a non-linear regression analysis using four-parameter sigmoidal equation using SigmaPlot software (v13 and v14.5).

## Cell culture

MDCK (kind gift of Dr Jacqueline Jouanneau, Institut Curie) and human cervical cancer HeLa cells, purchased from the American Type Culture Collection, were cultured in DMEM medium, supplemented with 10% FBS (Gibco, Life technologies) and 1/100 of ZellShieldTM (Minerva Biolabs GmbH). Twenty-four hours before treatment, the medium was exchanged with DMEM containing 0.1% FBS, and cells were then treated for subsequent experiments. SKOV-3 cells were cultured in RPMI 1640 medium supplemented with 10% FBS and penicillin–streptomycin all purchased from Gibco/Thermo Fisher Scientific.

## Western blots

The assay was performed according to Simonneau et al (2015). Hela cells were collected by scraping and lysed on ice with a lysis buffer (20 mM Hepes [pH 7.4], 142 mM KCl, 5 mM MgCl$_2$, 1 mM EDTA, 5% glycerol, 1% NP40, and 0.1% SDS) supplemented with freshly added protease inhibitor (1/200 dilution, #P8340; Sigma-Aldrich) and phosphatase inhibitor (1/400 dilution, #P5726; Sigma-Aldrich). Lysates were clarified by centrifugation (20,000$g$, 15 min), and protein concentration was determined (BCA protein assay Kit, Pierce, Thermo Fisher Scientific). The same protein amount (usually 20–30 $\mu$g) of cell extracts was separated by NuPAGE gel electrophoresis (4–12% Midi 1.0 mm Bis-Tris precast gels, Life technologies) and electro-transferred to polyvinylidene difluoride membranes (Merck Millipore). The membrane was cut between the 80- and 110-kD marker and at 50 kD to probe simultaneously phospho- or total MET, Akt, and ERK. Membranes were probed overnight at 4°C with primary antibodies diluted to 1/2,000 in 5% BSA and 0.1% sodium azide in PBS using specific total MET (#37-0100; Invitrogen), total ERK2 (#SC-154; Tebu-bio), phospho-MET (Y1234/1235, clone CD26, #3077; Cell Signaling), phospho-Akt (S473, clone CD9E, #4060; Cell Signaling), phospho-ERK (T202/Y204, clone E10, #9106; Cell Signaling). After extensive washing with PBS–0.05% Tween 20, incubation was followed with anti-mouse (#115-035-146) or anti-rabbit (#711-035-152) peroxidase-conjugated IgG secondary antibodies (Jackson ImmunoResearch) diluted to 1/30,000 in PBS-casein 0.2%. Protein–antibody complexes were visualized by chemiluminescence with the SuperSignal West Dura Extended Duration Substrate (Thermo Fisher Scientific) using a LAS-4000 imaging system (GE HeathCare Life Sciences) or X-ray films (CL-XposureTM Film; Thermo Fisher Scientific).

## AlphaScreen phosphorylation assay

Measurement of ERK1/2 (T202/Y204) and Akt1/2/3 (S473) activation were performed on HeLa cells using AlphaScreen SureFire Kits (#TGRES500 and #TGRA4S500) according to manufacturer protocol with minor modifications. Into a 96-well plate (#353072, Flacon, Corning), 10,000 HeLa were seeded in 200 $\mu$l of DMEM supplemented with 10% FBS and let attached and grown for 24 h. Cells were next starved in DMEM supplemented with 0.1% FBS for 2 h. Cells were treated for 8 min with MET agonist from 0.3 to 500 pM with HGF/SF, from 3 pM to 50 nM with K1K1 or K1K1H6, and from 3 to 1,000 nM with K1H6. Cells were quickly washed in cold PBS and AlphaScreen lysis buffer was added for 15 min under middle shaking (600 rpm on orbital rocker, MixMate, Eppendorf). Immediately, 5 $\mu$l of lysate in analysed by addition of acceptor and donor bead mixes for 2 h at 23°C. The emitted signal intensity was measured using standard Alpha settings on an EnSpire Multimode Plate Reader. Measured are expressed has experimental duplicates (mean ± SD, n = 2). The data were subjected to a non-linear regression analysis using four-parameter sigmoidal equation using SigmaPlot software (v13).

## SPR analysis

Affinity constants were measured using a Biacore T200 (GE Healthcare) at 25°C using PBST (PBS + 0.05% Tween 20) as running buffer and a flow rate of 30 $\mu$l/min. For the measuring of affinity for MET, the receptor fragment MET567 comprising the N-terminal SEMA domain and the cysteine-rich/PSI domain was immobilized at low-density (515 RU) using amine-coupling on a CM5 sensor chip (GE Healthcare). A blank inactivated reference channel was used to correct for non-specific binding. The 300-s injections were done using serial-diluted samples of K1K1, NK1, K1K1S2, and K1K1S4 testing a high range of 2–0.125 $\mu$M and low range of 0.8 $\mu$M–12.5 nM injecting for 300 s to reach equilibrium. For the measurement of affinity for heparin, a heparin-coated SC HEP0320.a chip (Xantec) was used. With no reference channel available, all four channels were used for analysis. K1K1, NK1, K1K1S2, and K1K1S4 were injected in a wide range of concentrations (10 $\mu$M–156 nM) for 300 s. Biacore Evaluation software version 3.2 was used for calculating affinity constants. Curves were plotted using GraphPad Prism (v5.04).

## SKOV-3 cell migration assays

Agonist-induced cell migration was measured by seeding 50,000 SKOV-3 cells resuspended in serum-free RPMI supplemented with 0.1% BSA (BioFroxx) in each of the upper wells of a 96-well Boyden chamber (AC96, Neuro Probe). HGF/SF, K1K1, K1K1S2, and K1K1S4 was loaded in the lower compartment, and cells were left to migrate in a humidified 37°C incubator with 5% CO$_2$ for 6 h. Afterwards, non-migrated cells were removed, and the cells migrated over the collagen-coated (100 $\mu$g/ml Purecol; Nutacon) 8-$\mu$m-pore–sized membrane (Neuro Probe) were fixed in 4% paraformaldehyde in PBS and stained using HCS CellMask Green stain (Thermo Fisher Scientific). Fluorescent intensity was imaged using an Azure C600 (Azure Biosystems), and migration was quantified by image analysis using the AzureSpot Pro analysis software after background-adjustment and charted in GraphPad Prism. Three independent

assays were performed with each condition tested in five wells ("quintuplicates"). To compare three independent assays, maximum HGF/SF migration observed at a concentration of 0.1 nM was set to 100%, and all other conditions were expressed as percentage thereof. Statistical significance was calculated with ANOVA followed by Dunnett's test with $P < 0.05$.

## MDCK scatter assay

The assay was performed according to Stoker et al (1987). MDCK cells were seeded at low density (2,000 cells/well on a 12-well plate, #353043 Falcon; Corning) in DMEM supplemented with 10% FCS to form compact colonies. After treatment, colony dispersion was observed, and the cells were fixed and coloured by the Hemacolor stain (Merck) according to the manufacturer's instructions. Representative images were captured using a phase-contrast microscope with 40× and 100× magnification (Nikon Eclipse TS100).

## Morphogenesis assay

Wells of a black thin clear bottom 24-well plate (#4ti-0241; Brooks Life Sciences) were coated with 200 µl of 2 mg/ml type 1 rat tail collagen (354249; Corning/BD Biosciences) diluted in DMEM equilibrated in 1/10 vol/vol of 7.5% wt/vol sodium bicarbonate solution (#25080-060, Gibco, Thermo Fisher Scientific). Cells (500–700) were seeded into a 500-µl layer of a 1:1 collagen/Growth Factor Reduced Matrigel (#354230; Corning/BD Biosciences) mixed in DMEM balance with bicarbonate and were treated twice a week for 1 mo. The cells were then fixed for 24 h with 4% paraformaldehyde in PBS, coloured for 3 h with 0.1 mg/l Evans Blue in PBS (#E2129; Sigma-Aldrich), extensively washed in PBS, and finally nuclei stained with 3 nM DAPI is PBS. Cell were observed using inverted microscope observed and confocal microscope (LSM880 Zeiss) with long distance 10× objective using 488 nm excitation Argon laser and LP 500–750 nm emission filter. 3D reconstitutions (14 µm slices) and maximum intensity projections of z-stack were realized with Zen software (v8.1).

## Survival assay

In a clear bottom 96-well plate (#353072, Flacon, Corning), 12,500 MDCK cells were seeded and incubated for 24 h in DMEM supplemented with 10% FBS. The culture media was next added with increasing concentrations of the HGF/SF, K1K1, K1K1H6, and NK1 agonists together with the apoptosis inducer anisomycin (1.4 µM) for 15 h. The cells were then washed with PBS to eliminate dead cells and then incubated for 1 h in 200 µl of HBSS (#14025092, Gibco, Thermo Fisher Scientific) containing 100 µM of Resazurin (#B21187; Sigma-Aldrich). Fluorescence was then measured with Envision multimode reader (Ex: 560 nm, Em: 590 nm) with maximal signal gain set (90%) on control cells wells and on top/top reading configuration. The data were normalized to percentage of control cells signal and subjected to a non-linear regression analysis using four-parameter sigmoidal equation with minimum = 0 and maximum = 100 using SigmaPlot software (v13 and v14.5).

## In vivo MET activation in the liver

All experimental procedures were conducted with the approval of the Ethics Committee for Animal Experimentation of the Nord Pas de Calais Region (CEEA 75). For kinetic analysis, 8-wk-old FVB mice weighing 19–21 g (Charles River) were i.p. injected with K1K1 (5 µg/mouse) and euthanized after 10, 30, 60, or 90 min of treatment. For dose–response analysis, 8-wk-old FVB mice weighing 19–21 g (Charles River) were i.p. injected with 0.1, 0.5, 1, and 5 µg of K1K1 diluted in 100 µl of PBS and euthanized per cervical dislocation 10 min after injection. To analyse the effect of the route of administration, 12-wk-old C57BL/6 NRJ mice weighing 19–21 g (Janvier Labs) were i.p. or i.v. injected with K1K1 (5 µg/mouse) and euthanized after 10 min of treatment. Livers were immediately perfused with PBS supplemented with protease and phosphatase inhibitors, collected, and then rapidly frozen in liquid nitrogen for subsequent protein extraction and Western blot analysis.

## Liver steatosis model

Eighty C57BL/6J female mice of 8–10 wk (~19 g) provided by Janvier Labs were used in this study. They were housed in temperature- and humidity-controlled rooms, kept on a 12-h light–dark cycle. Animal procedures were conducted in accordance with French government policies. Mice were fed with a control liquid diet of adapted Lieber DeCarli ad libitum during a 7-d habituation process. Body weight and liquid diet consumption were monitored every 2 d. Mice were then separated in six groups:

  (i) LDC control diet–fed mice + vehicle (n = 10, vehicle: 0.0 µg)
 (ii) LDC control diet–fed mice + product (n = 10, highest dose: 10 µg)
(iii) LDC OH-fed mice + vehicle (n = 15, vehicle: 0.0 µg)
(iv) LDC OH-fed mice + product (n = 15, dose 1: 0.4 µg)
 (v) LDC OH-fed mice + product (n = 15, dose 2: 2.0 µg)
(vi) LDC OH-fed mice + product (n = 15, dose 3: 10 µg)

  During a 10-d alcoholization period, mice received five i.p. injections of 100 µl of sterile PBS or different doses of K1K1 diluted in sterile PBS. Ethanol-fed groups had unlimited access to adapted LDC containing ethanol and control mice were pair-fed with iso-caloric control diet over this feeding period. On the 11th day, mice were euthanized.

## Hematoxylin-erythrosine B staining

The livers embedded in paraffin and sectioned at 4 µm thickness were deparaffinized and rehydrated in successive baths of xylene, absolute ethanol, ethanol 95%, and tap water. The slides were stained with hematoxylin (Harris), rinsed, and stained with erythrosine B 0.8% (Sigma-Aldrich). The slides were finally dehydrated and mounted with Eukitt (GmbH). Slides were analysed on a Zeiss Axio Imager M2 microscope. For histological evaluation, a score between 0 and 3 (3 = 75% steatosis) for steatosis was assigned to each section of the liver (3 areas per section).

## Real-time quantitative RT–PCR

Livers in 350 $\mu$l of RA1 and 3.5 $\mu$l of $\beta$-mercaptoethanol were mixed and frozen at –80°C until RNA extraction. RNA extraction was carried out using the nucleospin RNA kit (Macherey-Nagel). The RNAs were quantified using a NanoDrop, and the retro-transcription made using the high-capacity cDNA reverse-transcription kit (Thermo Fisher Scientific). qPCRs were done by mixing 3.25 $\mu$l of $H_2O$ RNAse free from the RNA extraction kit to 6.25 $\mu$l of Fast SYBR green Master mix (Thermo Fisher Scientific) and 0.25 $\mu$l of reverse and 0.25 $\mu$l of forward primers diluted in 1/10th in distilled water. qPCRs were performed in a StepOne plus ther-mocycler (Thermo Fisher Scientific). The sequences of the primers used in the study are:

$\beta$-actin F: CCATGTCGTCCCAGTTGGTAA
$\beta$-actin R: GAATGGGTCAGAAGGACTCCTATGT
IL-6 F: GTTCTCTGGGAAATCGTGGA
IL-6 R: CAGAATTGCCATTGCACAAC
TNF$\alpha$ F: TGGGAGTAGACAAGGTACAACCC
TNF$\alpha$ R: CATCTTCTCAAAATTCGAGTGACAA
MET F: GACAAGACCACCGAGGATG
MET R: GGAGCCTTCATTGTGAGGAG
ApoB F: TACTTCCACCCACAGTCCC
ApoB R: GGAGCCTAGCAATCTGGAAG
LDLR F: TTCAGTGCCAATCGACTCAC
LDLR R: TCACACCAGTTCACCCCTCT
PPAR$\alpha$ F: AGGCAGATGACCTGGAAAGTC
PPAR$\alpha$ R: ATGCGTGAACTCCGTAGTGG

Results were presented as levels of expression relative to that of controls after normalizing with $\beta$-actin mRNA using the compar-ative Ct method.

# Data Availability

Structural dataset for K1K1 and K1K1H6 are available from the PDB data bank with id code 7OCL and 7OCM, respectively. The SAXS data are available from the SASBDB database with id codes SASDLM9, SASDLN9, and SASDLP9.

Complete source datasets (.zip and .pdf files) have been uploaded together with the supplementary materials for each figure.

# Supplementary Information

# Acknowledgements

We thank SATT-Nord (France) for financing the proof-of-concept experi-ments and Thierry Chassat from the PLETHA animal facility in the Lille Pasteur Institute for helpful advises and kind availability. Work in the laboratory in Pavia was made possible through funding from the Italian Ministry for Universities and Research (Departments of Excellence initiative) and from the Leukemia Research Fund (AIL Trentino). We would like to acknowledge Dr Maristella Magi (University of Pavia) for help at the ESRF beamline, Dr Claudia Scotti (University of Pavia), and Dr Dimitri Y Chirgadze (University of Cambridge, UK) for help with structural analysis of K1K1 and K1K1H6 and Maria Cristina Barbieri for technical support and Dr Raymond Pierce for proofreading.

## Author Contributions

G de Nola: conceptualization, formal analysis, visualization, methodology, and writing—original draft.
B Leclercq: conceptualization, data curation, formal analysis, visuali-zation, methodology, and writing—original draft, review, and editing.
A Mougel: supervision, visualization, methodology, and wri-ting—original draft.
S Taront: formal analysis, visualization, methodology, and wri-ting—original draft.
C Simonneau: conceptualization, visualization, and methodology.
F Forneris: conceptualization, data curation, visualization, and methodology.
E Adriaenssens: conceptualization, formal analysis, validation, vi-sualization, methodology, and writing—original draft.
H Drobecq: visualization and methodology.
L Iamele: conceptualization, supervision, visualization, and methodology.
L Dubuquoy: conceptualization, supervision, visualization, meth-odology, and writing—original draft.
O Melnyk: conceptualization, resources, data curation, formal analysis, supervision, funding acquisition, validation, investigation, visualization, project administration, and writing—original draft, review, and editing.
E Gherardi: conceptualization, resources, formal analysis, super-vision, funding acquisition, investigation, visualization, methodol-ogy, and writing—review and editing.
H de Jonge: conceptualization, resources, data curation, formal analysis, supervision, funding acquisition, validation, investigation, visualization, methodology, project administration, and wri-ting—original draft, review, and editing.
J Vicogne: conceptualization, resources, data curation, formal analysis, supervision, funding acquisition, validation, investigation, visualization, methodology, project administration, and writing—original draft, review, and editing.

## Conflict of Interest Statement

The authors declare no competing interests. The Universities of Lille and Pavia have filed a joint patent on K1K1 (patent number WO2016116578A1).

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
