## [Reviewer comments · Life Science Alliance]

Life Science Alliance

Dimerization of kringle 1 domain from HGF/SF provides a potent minimal MET receptor agonist

Giovanni De Nola, Bérénice Leclercq, Alexandra Mougel, Solenne Taront, Claire Simonneau, Federico Forneris, Eric Adriaenssens, Herve Drobecq, Luisa Iamele, Laurent Dubuquoy, Oleg Melnyk, Ermanno Gherardi, Hugo de Jonge, and Jérôme Vicogne

DOI: <https://doi.org/10.26508/lsa.202201424>

Corresponding author(s): Jérôme Vicogne, University of Lille, CNRS, Inserm, CHU Lille, Institut Pasteur de Lille, U1019 - UMR 9017 - CILL - Center for Infection and Immunity of Lille and Hugo de Jonge, University of Pavia

Review Timeline:

Submission Date:	2022-02-25
Editorial Decision:	2022-04-19
Revision Received:	2022-06-20
Editorial Decision:	2022-06-27
Revision Received:	2022-07-01
Accepted:	2022-07-06

Scientific Editor: Novella Guidi

Transaction Report:

April 19, 2022

Re: Life Science Alliance manuscript #LSA-2022-01424-T

Dr. Jérôme Vicogne
University of Lille, CNRS, Inserm, CHU Lille, Institut Pasteur de Lille, U1019 - UMR 9017 - CIIL - Center for Infection and Immunity of Lille
1 rue du Pr Calmette
Lille 59021
France

Dear Dr. Vicogne,

Thank you for submitting your manuscript entitled "Dimerization of kringle 1 domain from HGF/SF provides a potent minimal MET receptor agonist" to Life Science Alliance. The manuscript was assessed by expert reviewers, whose comments are appended to this letter. We invite you to submit a revised manuscript addressing the Reviewer comments.

Thank you for this interesting contribution to Life Science Alliance. We are looking forward to receiving your revised manuscript.

Sincerely,

B. MANUSCRIPT ORGANIZATION AND FORMATTING:

Reviewer #1 (Comments to the Authors (Required)):

The paper of de Nola et al is a very interesting study, which describes the development of a novel MET agonist, namely K1K1, derived from the natural HGF ligand. The HGF is a well-recognized promoter of cell migration and tissue regeneration, thereby the described recombinant molecule could potentially represent a tool for translational medicine applications. I am not able to review and discuss the structural data of K1K1 and the proposed mechanism of 1:2 K1K1: MET configuration. However, I find very convincing the data showing that the new molecule has 1) low affinity binding for heparin and 2) high potency on MET activation. The in vivo efficacy of K1K1 highlights the potentially promising perspective of this new molecule.

The manuscript is well written, technically sound and the results are very convincing. The experiments are carefully performed and the results are clearly presented. It is well-grounded on the literature, which is well referenced. I would recommend the publication of this paper essentially as it is.

Reviewer #2 (Comments to the Authors (Required)):

In this manuscript, a novel agonist of HGF (Hepatocyte Growth Factor) and its characterisation is reported. It consists in a dimer of the first Kringle domain of HGF, called K1K1. The rationale for its design and the determination of its structure are presented. Its role in binding MET and in in vitro functional assays and in vivo (as treatment of alcoholic steatohepatitis) was determined. In vitro, comparisons were made with HGF and K1K1 mutated in its the low-affinity heparin binding sites (K1K1S2 and K1K1S4). It was concluded that K1K1 is a potent agonist of HGF, suggesting it could be used for therapeutic purpose in the future for various diseases in the liver or other diseases such as COPD.

This is an interesting manuscript with potential important clinical implication. The paper is clearly written and presented and experiments appear well performed.

Minor comments:

Figure 5B: The difference in 3D morphogenesis between 10nM of K1K1S2 and 10nM of K1K1 are not clear. It seems that K1K1S2 is still able to stimulate the 3D morphogenesis. Is there a way to provide quantitative data of this assay?

Figure 5C: It would be good to obtain statistical results from at least 3 independent experiments in this assay as the scattering and the morphogenesis assay are not quantitative.

It would be important to discuss the possible adverse effects of treating patients with such a MET agonist, such as promotion of cancer development or progression.

Dear editor,

We thank the reviewers for their very positive evaluations of our work and we are pleased to propose to LSA a revised version of our manuscript taking into consideration the comments from Reviewer#2. In particular we have added a quantitative assay (Fig. 5C). All modification/addition have been highlighted within the text.

- *Figure 5B: The difference in 3D morphogenesis between 10 nM of K1K1S2 and 10 nM of K1K1 are not clear. It seems that K1K1S2 is still able to stimulate the 3D morphogenesis. Is there a way to provide quantitative data of this assay?*

In this morphogenesis assay, we have selected the “most branched” observable structure for each condition. Indeed, it is very difficult to make quantification in a thick Matrigel layer since only a small volume of the well is accessible for imaging. In addition, in agreement with the scattering (Fig. 5A) and migration assays (Fig. 5C), K1K1S2 at doses >10 nM still induced a significant tubulogenesis, even if less pronounced than HGF/SF and K1K1, whereas the K1K1S4 is ineffective with a net proliferation (spheres) but no tubulogenesis. This point has been added in the results section.

- *Figure 5C: It would be good to obtain statistical results from at least 3 independent experiments in this assay as the scattering and the morphogenesis assay are not quantitative.*

We have performed three independent experiments with each condition present five times and averaged. To allow comparison between the independent assays, we made the migration induced by 0.1 nM HGF/SF - which is the maximum observed in each assay - 100% and used this to calculate the percentage of migration for all other conditions. For statistical analysis, we used the Anova Dunnett's test and marked the conditions that significantly differed with an asterisk.

- It would be important to discuss the possible adverse effects of treating patients with such a MET agonist, such as promotion of cancer development or progression.

We have added this point at the end of the discussion section with the corresponding references.

The manuscript and figures have been edited to fit the LSA guidelines. All source data have been collected and uploaded together with supplementary information.

June 27, 2022

RE: Life Science Alliance Manuscript #LSA-2022-01424-TR

Dr. Jérôme Vicogne
University of Lille, CNRS, Inserm, CHU Lille, Institut Pasteur de Lille, U1019 - UMR 9017 - CIIL - Center for Infection and Immunity of Lille
1 rue du Pr Calmette
Lille 59021
France

Dear Dr. Vicogne,

Thank you for submitting your revised manuscript entitled "Dimerization of kringle 1 domain from HGF/SF provides a potent minimal MET receptor agonist". We would be happy to publish your paper in Life Science Alliance pending final revisions necessary to meet our formatting guidelines.

- please add ORCID ID for secondary corresponding author-they should have received instructions on how to do so
- please add a conflict of interest statement to the main manuscript text
- please consult our manuscript preparation guidelines <https://www.life-science-alliance.org/manuscript-prep> and make sure your manuscript sections are in the correct order
- please add your supplementary figure and table legends to the main manuscript text and provide your table files in editable doc or excel file format
- please add a callout for Figure 2A & C; Figure 3B; and Figure 4C to your main manuscript text
- please upload your main figures as single files; these will be displayed in-line in the HTML version of your paper, so please provide them as single page files (Figure 1 currently spans 2 pages); we do not have a limit on the number of main figures and these can be split if necessary for space
- 3 secondary Corresponding Authors are listed; it is LSA's policy to limit the number of Corresponding Authors to 2

Figure Check:

- The scale bars in Figures 7, S4 and S7 are hard to see

A. FINAL FILES:

B. MANUSCRIPT ORGANIZATION AND FORMATTING:

Sincerely,

Reviewer #2 (Comments to the Authors (Required)):

I am satisfied with the revised manuscript and recommend publication

July 6, 2022

RE: Life Science Alliance Manuscript #LSA-2022-01424-TRR

Dr. Jérôme Vicogne
University of Lille, CNRS, Inserm, CHU Lille, Institut Pasteur de Lille, U1019 - UMR 9017 - CIIL - Center for Infection and Immunity of Lille
1 rue du Pr Calmette
Lille 59021
France

Dear Dr. Vicogne,

Thank you for submitting your Research Article entitled "Dimerization of kringle 1 domain from HGF/SF provides a potent minimal MET receptor agonist". It is a pleasure to let you know that your manuscript is now accepted for publication in Life Science Alliance. Congratulations on this interesting work.

DISTRIBUTION OF MATERIALS:

Again, congratulations on a very nice paper. I hope you found the review process to be constructive and are pleased with how the manuscript was handled editorially. We look forward to future exciting submissions from your lab.

Sincerely,
